# MODALITY-AGNOSTIC FMRI DECODING OF VISION AND LANGUAGE

**Mitja Nikolaus**[*]
CerCo, CNRS
Toulouse, France
mitja.nikolaus@cnrs.fr

**Milad Mozafari**[*]
Torus AI
Toulouse, France
milad.mozafari@torus-actions.fr

**Nicholas Asher**
IRIT, Université Paul Sabatier
Toulouse, France
nicholas.asher@irit.fr

**Leila Reddy & Rufin VanRullen**
CerCo, CNRS
Toulouse, France
{leila.reddy,rufin.vanrullen}@cnrs.fr

## ABSTRACT

Previous studies have shown that it is possible to map brain activation data of subjects viewing images onto the feature representation space of not only *vision* models (modality-specific decoding) but also *language* models (cross-modal decoding). In this work, we introduce and use a new large-scale fMRI dataset ($\sim 8,500$ trials per subject) of people watching both images and text descriptions of such images. This novel dataset enables the development of *modality-agnostic* decoders: a single decoder that can predict which stimulus a subject is seeing, irrespective of the modality (image or text) in which the stimulus is presented. We train and evaluate such decoders to map brain signals onto stimulus representations from a large range of publicly available vision, language and multimodal (vision+language) models. Our findings reveal that (1) modality-agnostic decoders perform as well as (and sometimes even better than) modality-specific decoders (2) modality-agnostic decoders mapping brain data onto representations from unimodal models perform as well as decoders relying on multimodal representations (3) while language and low-level visual (occipital) brain regions are best at decoding text and image stimuli, respectively, high-level visual (temporal) regions perform well on both stimulus types.

## 1 INTRODUCTION

Advances in deep-learning-based computational models of language and vision paired with large-scale open source fMRI datasets have fostered the development of brain decoding models which classify or reconstruct stimuli that subjects were seeing based on their brain activations (Naselaris et al., 2009; Nishimoto et al., 2011; Pereira et al., 2018; VanRullen & Reddy, 2019; Ozcelik & VanRullen, 2023; Scotti et al., 2023; Tang et al., 2023b; Benchetrit et al., 2023; Karamolegkou et al., 2023; Xia, 2024). A range of studies have presented *modality-specific* mappings between fMRI brain activation data of subjects viewing stimuli in one modality (e.g. images) and feature representation space of models of the same modality (e.g. vision models). More recently, it has been shown that these mappings can also be trained in a cross-modal fashion, i.e. mappings between fMRI data from one modality and feature space of models of another modality (e.g. between fMRI data of subjects viewing images and representations from language models) (Matsuo et al., 2017; Takada et al., 2020; Huang et al., 2021; Ferrante et al., 2023; Tang et al., 2023a).

Here, we present a new fMRI dataset and use it to develop *modality-agnostic* decoders. A modality-agnostic decoder is trained on fMRI data from multiple modalities (here: vision and language) and can retrieve the stimulus (image or caption) a subject is seeing irrespective of the modality. In contrast to *modality-specific* decoders that can be applied only in the single modality that they were

---

[*]Joint first authors

trained on, *modality-agnostic* decoders can be applied in multiple modalities, even without knowing the stimulus modality a priori.

The fMRI experiment consists of 6 subjects viewing $\sim 8,500$ stimuli (images and captions) while performing a one-back cross-modal matching task. An additional set of 70 images and 70 captions were presented to all subjects and serves as a test set for the decoders. This new fMRI dataset will be released publicly in an upcoming publication.

We train modality-agnostic decoders based on this new multimodal fMRI dataset and evaluate them on their decoding performance in both modalities. Our results show that modality-agnostic decoders generally perform on par with their respective modality-specific counterparts, despite the additional challenge of uncertainty about the stimulus modality. We further compare decoders trained on features extracted from a range of vision, language and multimodal models and show that multimodal representations do not increase decoding performance above that of decoders trained on unimodal representations in the correct modality. Finally, an ROI-based analysis reveals that activity from high-level visual brain regions is most effective for training modality-agnostic decoders, suggesting that these regions contain representations that are to some degree "amodal".

## 2 METHODS

### 2.1 FMRI EXPERIMENT

Six subjects (2 female, age between 20 and 50 years, all right-handed) participated in the experiment after providing informed consent. The study was performed in accordance with French national ethical regulations (Comité de Protection des Personnes, ID 2019-A01920-57). We collected functional MRI data using a 3T Philips ACHIEVA scanner. At the start of each session, we further acquired high-resolution anatomical images for each subject. Scanning was spanned over 10 sessions (except for sub-01: 11 sessions), each consisting of 13-16 runs during which the subjects were presented 86 stimuli. The stimulus type varied randomly between images and captions. Each stimulus was presented for 2.5 seconds at the center of the screen, the inter-stimulus interval was 1s. Further details on the scanner configuration and experimental setup are reported in Appendix A.1.

Subjects were performing a one-back matching task: They were instructed to press a button whenever the stimulus was matching the immediately preceding one. In case the previous stimulus was of the same modality (e.g. two captions in a row), the subjects were instructed to press a button if the stimuli were matching exactly. In the cross-modal case (e.g. an image followed by a caption), the button had to be pressed if the caption was a valid description of the image, and vice versa. Positive one-back trials occurred on average every 10 stimuli.

Images and captions were taken from the training and validation sets of the COCO dataset (Lin et al., 2014, COCO contains 5 matching captions for each image, of which we only considered the shortest one in order to fit on the screen and to ensure a comparable length for all captions). As our training set, a random subset of images and another random subset of captions was selected for each subject. All these stimuli were presented only a single time. Additionally, a shared subset of 140 stimuli (70 images and 70 captions) was presented repeatedly (on average: 26 times, min: 22, max: 31) to each subject in order to reduce noise, serving as our test set. These stimuli were inserted randomly between the training stimuli. Note that for each image (respectively, caption) presented to the subject, we retained the corresponding caption (resp. image) in order to estimate model features in the opposite modality (e.g. language model features for an image stimulus).

### 2.2 FMRI PREPROCESSING

Preprocessing of the fMRI data was performed using SPM 12 (Ashburner et al., 2014). We applied Slice Time Correction and Realignment for each subject. Each session was coregistered with the subject's T1 scan. Afterwards, we transformed all data to the MNI305 space (Evans et al., 1993) using Freesurfer (Fischl, 2012), and explicit gray matter masks were created using SPM and applied for each subject.

We fit a first GLM for each subject on data from all sessions. We included regressors for events that re-occurred across runs and sessions, i.e. test images, test captions, fixations, blank screens,

and one-back trials. The residual volumes of these GLMs were the inputs to a second-phase GLM, which was fit for each run separately, and intended to derive single-trial beta-values. We included regressors for each training image and caption presented during the run. As output of these second GLMs we obtained a single volume of beta-values for each training caption and image. One-back target trials were excluded from the second-phase GLM.

## 2.3 MODALITY-AGNOSTIC DECODERS

We trained regression models that take fMRI beta-values from training stimuli (images and captions) as input and predict latent representations extracted from vision, language, and multimodal models.

We consider as vision models: ResNet (He et al., 2016), ViT (Dosovitskiy et al., 2020), and DINOv2 (Oquab et al., 2023); as language models: BERT (Devlin et al., 2019), GPT2 (Radford et al., 2019), Llama2 (Touvron et al., 2023), mistral and mixtral (Jiang et al., 2023). Regarding multimodal models, we extract features from VisualBERT (Li et al., 2019), BridgeTower (Xu et al., 2023), LXMERT (Tan & Bansal, 2019), ViLT (Kim et al., 2021), CLIP (Radford et al., 2021), ImageBind (Girdhar et al., 2023), Flava (Singh et al., 2022). For each target stimulus (image or caption), we extracted model features from the corresponding image for vision models, the corresponding caption for language models, and a concatenated representation of both image and caption for the multimodal models. We use publicly available pretrained models implemented in the HuggingFace Transformers library (Wolf et al., 2020). In order to estimate the effect of model training, we further extract features from a randomly initialized Flava model as a baseline. Further details on feature extraction and decoder training can be found in Appendix A.2.

The decoders were linear ridge-regression models implemented using scikit-learn (Pedregosa et al., 2011). The regularization hyperparameter $\alpha$ was optimized using 5-fold cross validation on the training set (values considered: $\alpha \in \{1e3, 1e4, 1e5, 1e6, 1e7\}$). Afterwards, a final model was trained using the best $\alpha$ on the whole training set.

Finally, the models were evaluated on the held-out test data (140 stimuli, 70 captions and 70 images) using pairwise accuracy calculated using cosine distance. In the case of cross-modal decoding (e.g. mapping an image stimulus into the latent space of a language model), a trial was counted as correct if the caption corresponding to the image (according to the ground-truth in COCO) was closest.

## 3 RESULTS

### 3.1 MODALITY-AGNOSTIC DECODING

We present a comparison of pairwise accuracy scores for captions and images of modality-agnostic and modality-specific decoders in Figure 1. Results for individual subjects can be found in Appendix A.5. Generally, we observe that modality-agnostic decoders perform as well as the modality-specific decoders trained on the correct modality, and much better than the modality-specific decoders trained on the opposite modality. They achieve this high performance despite the additional challenge of not knowing the modality of the stimulus the subject was seeing.

When calculating the overall modality-agnostic decoding performance as average performance for captions and images (bottom panel of Figure 1), we find that modality-agnostic decoders based on the best multimodal features (ViLT: $0.88 \pm 0.03$) do not perform substantially better than decoders based on the best language features (GPT2-xl: $0.86 \pm 0.03$) and only slightly better than decoders trained on the best vision features (Dino-large: $0.83 \pm 0.03$).

### 3.2 ROI-BASED MODALITY-AGNOSTIC DECODING

The results presented so far are based on decoders trained on data from the whole brain. To provide insight into the organization of visual and language representations in the brain, we additionally trained decoders on subsets of voxels for 3 Regions Of Interest (ROI), defined based on an anatomical atlas (Destrieux et al., 2010): A low-level visual area spanning mainly the occipital lobe, a high-level visual area in the temporal lobe, and a left-lateralized language-related area broadly defined based on the findings of Fedorenko et al. (2010). Surface plots of these 3 ROIs are depicted in Figure 2. Further details on the ROI definition can be found in Appendix A.4.

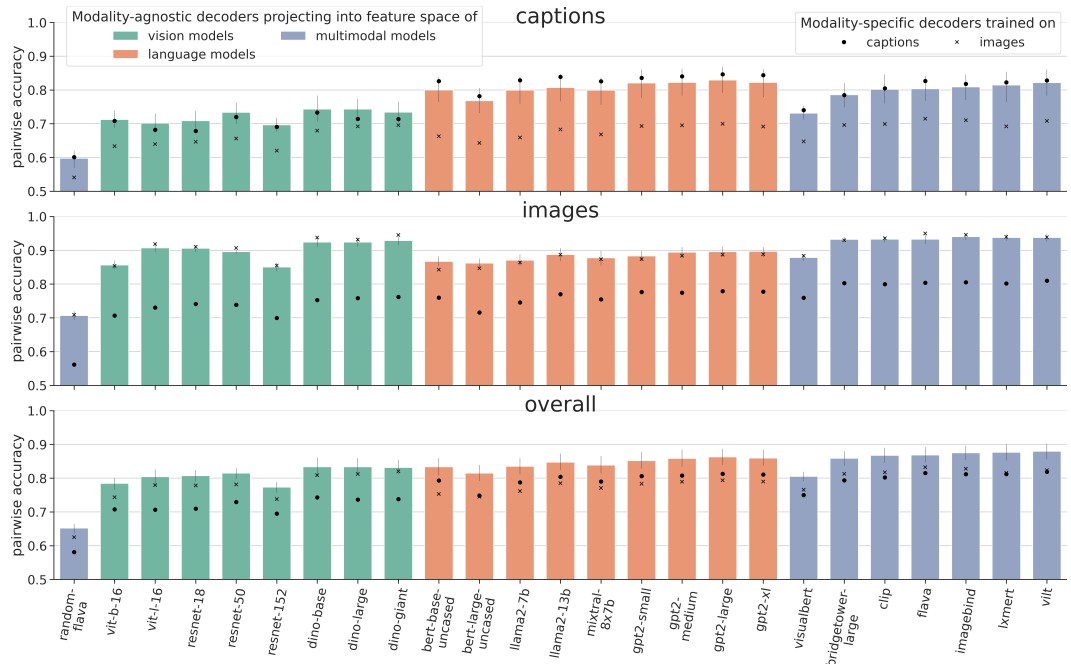

Figure 1: Decoding accuracy for captions (top), images (middle) and overall (bottom) for modality-agnostic decoders trained on full data (bars), compared to modality-specific decoders trained on either just linguistic fMRI data (•) or just on visual fMRI data (×). Error bars indicate 95% confidence intervals for modality-agnostic decoders. Chance performance is at 0.5.

Pairwise decoding accuracies for the ROI-based decoders are presented in Figure 3. Even though the ROI-based decoders rely on 20x less dimensions (∼10,000 voxels) than the whole brain decoders (∼215,000 voxels), image decoding performance of decoders based on the low-level visual ROI is on par with decoders that use the whole brain data, and caption decoding performance for the language ROI is close to it as well.

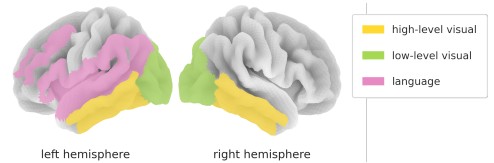

Figure 2: Surface plots of the 3 ROIs. The average numbers of voxels in the high-level visual area is 11,340; in the low-level visual area 10,578; and in the language area 11,193.

As expected, we find that both modality-agnostic and modality-specific decoders' decoding accuracy for captions is lowest in the low-level visual area and highest in the language area; for images, it is lowest in the language area and highest in the low-level visual area. However, decoders trained on high-level visual areas of the temporal cortex perform well, both for decoding images and captions, and are systematically the highest across both modalities (Fig 3, bottom). This suggests that representations in this area are to some degree amodal.

## 4 DISCUSSION

In this study, we presented a novel large-scale fMRI dataset and used it to train modality-agnostic decoders for vision and language. The fMRI data is unique in that it contains a large number of *separate* trials for *matched* visual and language stimuli (images and captions from the COCO dataset). Previous studies that relied on unimodal fMRI data (e.g. Chang et al., 2019; Allen et al., 2022) required either manual annotations to map stimuli from multiple modalities into a shared semantic space (Popham et al., 2021) or training of linear transformation matrices based on additional multimodal paired training data (Tang et al., 2023a). Other multimodal fMRI datasets usually consist of *simultaneous* presentations of visual and language stimuli (e.g. movies Huth et al., 2012; Çukur et al., 2013; Cichy & Lahner, 2021), which allows for the study of multimodal feature integration

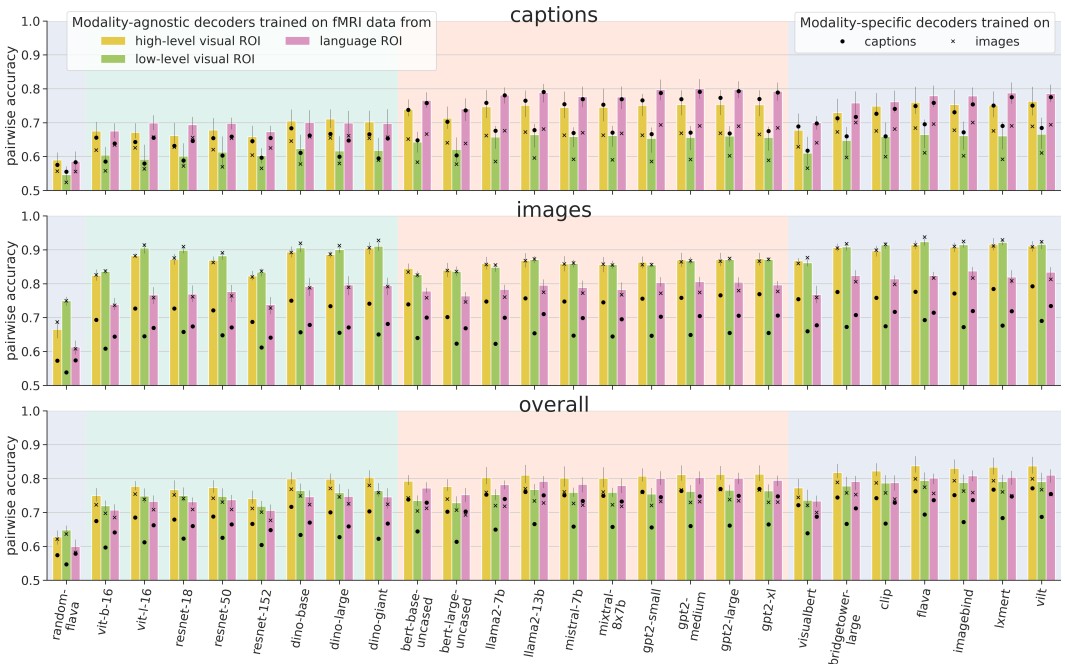

Figure 3: Decoding accuracy for captions (top), images (middle) and overall (bottom) for modality-agnostic (bars) and modality-specific decoders (• and ×) trained on 3 ROIs. The background colors reflect the model features that the decoders project into (vision, language or multimodal).

(Bonnici et al., 2016; Khosla et al., 2021; Dong & Toneva, 2023), but does not allow for the study of modalities in isolation.

The results of our decoding experiments based on this new dataset suggest that in order to build modality-agnostic decoders, we do not necessarily need representations from multimodal models; unimodal representations (especially from language models) can lead to comparably high performance. Two recent studies found that multimodal transformers (CLIP and BridgeTower) learn more aligned representations in language and vision than unimodal transformers (Wang et al., 2023; Tang et al., 2023a). In our study, we evaluated a large range of unimodal and multimodal representations, and found that especially representations extracted from more recent large language models (e.g. GPT2-xl) are as good as multimodal representations. Reasons for these different results could be that the aforementioned studies only considered language representations extracted from substantially smaller language models (BERT and RoBERTa (Liu et al., 2019)) and that models were compared in terms of their *encoding* performance, while we measured *decoding* performance (Kriegeskorte & Douglas, 2019).

Tang et al. (2023a) trained cross-modal encoding models between data from participants viewing movies and listening to audio books and found that "tuning for concepts in language and vision is positively correlated in most regions outside of visual cortex, it is negatively correlated in visual cortex." This phenomenon could explain why we do not observe higher performance of modality-agnostic decoders compared to modality-specific ones when trained on low-level visual ROIs: If the same stimuli presented in the visual and language modality are represented differently in these ROIs, training in one modality will not improve performance in the other modality.

Our ROIs contain several "amodal" regions that have been identified in previous studies (Devereux et al., 2013; Fairhall & Caramazza, 2013; Popham et al., 2021), such as the middle and inferior temporal gyrus (part of the high-level visual ROI) and the left angular gyrus and left posterior cingulate gyrus (language ROI). The superior performance of modality-agnostic decoders for these ROIs confirms that these regions share representations between modalities. In future work, we plan to perform a more fine-grained searchlight-based analysis to identify specific "amodal" regions, i.e. regions in which the performance advantage of modality-agnostic decoders is highest.

ACKNOWLEDGMENTS

This research was funded by grants from the French Agence Nationale de la Recherche (ANR: AI-REPS grant number ANR-18-CE37-0007-01 and ANITI grant number ANR-19-PI3A-0004) as well as the European Union (ERC Advanced grant GLoW, 101096017). Views and opinions expressed are however those of the author(s) only and do not necessarily reflect those of the European Union or the European Research Council Executive Agency. Neither the European Union nor the granting authority can be held responsible for them.

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

# A  APPENDIX

## A.1  FMRI EXPERIMENT DETAILS

The functional MRI data was collected using a 3T Philips ACHIEVA scanner (gradient echo pulse sequence, TR=2s, TE=10ms, 41 slices with a 32-channel head coil, slice thickness=3mm with 0.2mm gap, in-plane voxel dimensions 3×3mm). High-resolution anatomical images for each subject (1×1×1mm voxels, TR=8.13ms, TE=3.74ms, 170 sagittal slices) were acquired at the start of each session.

Each run started and ended with an 8s fixation period. The stimulus type varied randomly between images and captions. Each stimulus was presented for 2.5 seconds at the center of the screen (visual angle: 14.6 degrees), captions were displayed in white on a gray background (font: "Consolas"). The inter-stimulus interval was 1s. Every 10 stimuli there was a fixation trial that lasted for 2.5s. Every 5min there was a longer fixation trial for 16s.

Exact numbers of training stimuli presented for each subject can be found in Table 1.

Table 1: Number of training stimuli for each subject.

| Subject | # Stimuli |
|---------|-----------|
| sub-01  | 9856      |
| sub-02  | 8232      |
| sub-03  | 8008      |
| sub-04  | 8680      |
| sub-05  | 8568      |
| sub-06  | 8568      |

## A.2  FEATURE EXTRACTION DETAILS

Pretrained models were taken from Huggingface or from their respective authors' repositories. Model versions for unimodal models are as indicated in Figure 1. For multimodal models, the exact version for CLIP was `clip-vit-large-patch14`, for ViLT `vilt-b32-mlm`, for LXMERT `lxmert-base-uncased`, for VisualBERT `visualbert-nlvr2-coco-pre`, for Imagebind `imagebind_huge`, and for Flava `flava-full`.

We extracted language features from all models by averaging the outputs for each token, as this has established as common practice for the extraction of sentence embeddings from Transformer-based language models (e.g. Krasnowska-Kieraś & Wróblewska, 2019; Reimers & Gurevych, 2019).

For Transformer-based vision models, we compare representations extracted by averaging the outputs for each patch with representations extracted from `[CLS]` tokens in Figure 4. We find that for almost all models, the mean features allow for higher decoding accuracies. For all experiments reported in the main paper we therefore only considered this method.

For multimodal models, we concatenated the vision and language features to create the final multimodal feature representation. We also trained decoders on only the language or vision features of the multimodal models. Their performance was in most cases comparable or worse than for the concatenated features, therefore we do not report them in the main text. Results using these features can however be found in Appendix A.3.

The models Flava and BridgeTower also allow for a direct extraction of multimodal features, we found however that they perform much worse than concatenated vision and language features and therefore did not consider these further in our experiments.

## A.3  RESULTS FOR VISION AND LANGUAGE FEATURES OF MULTIMODAL MODELS

In the main results, we only consider concatenated vision and language features of the multimodal models. We can however also just use the vision or language features of these models.

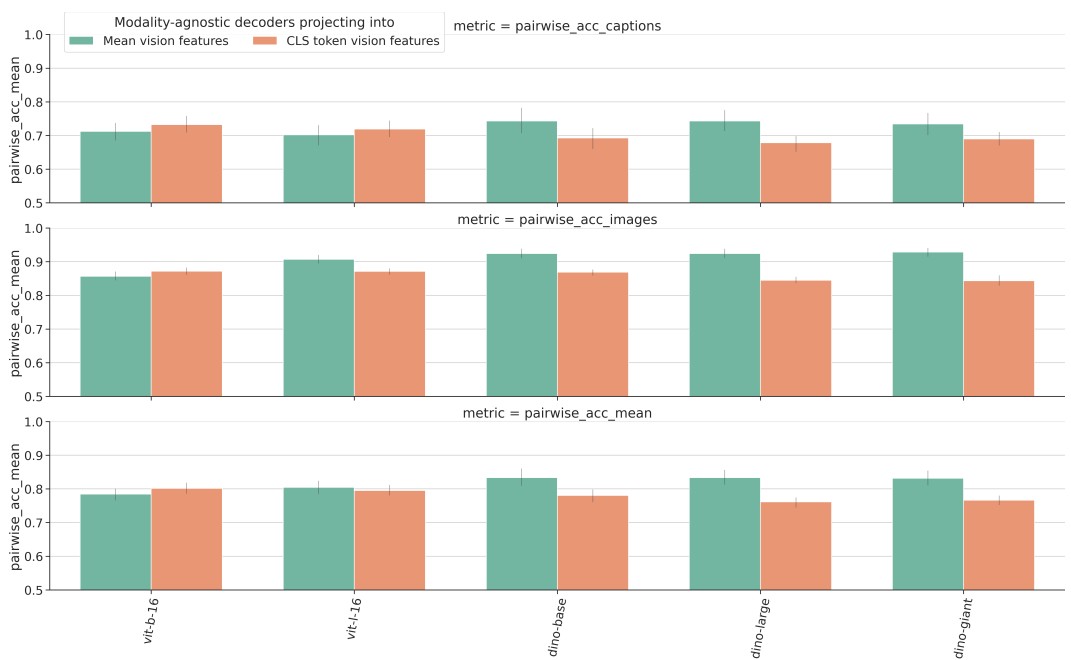

Figure 4: Pairwise accuracy for decoders based on vision features extracted by averaging the last hidden states ("Mean vision features") compared to when using features extracted from `[CLS]` tokens.

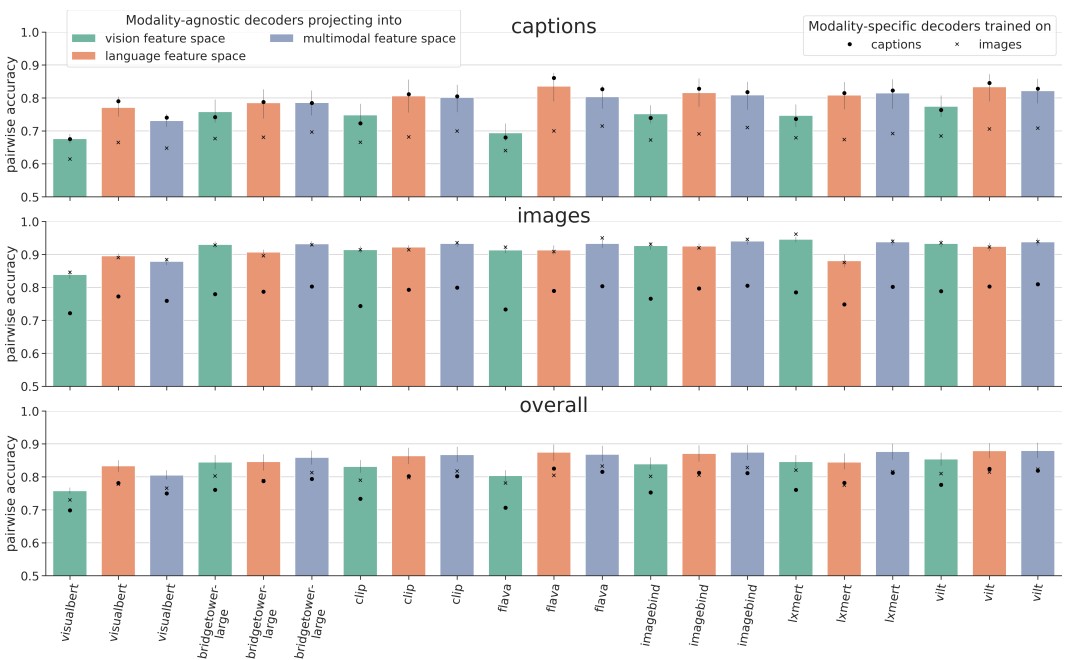

Figure 5: Pairwise accuracy for decoders based on vision, language, and multimodal features extracted from multimodal models.

Figure 5 compares the performance of decoders based on these different features for multimodal models. We find that the concatenated multimodal and language features usually perform best, in line with the main results comparing unimodal and multimodal features in Figure 1.

## A.4 ROI DETAILS

We defined ROIs based on the anatomical Destrieux Atlas (Destrieux et al., 2010). The exact labels and names for each region included in each ROI can be found in Tables 2, 3, and 4. We defined non-overlapping regions of comparable size (in terms of number of voxels). While these ROI definitions allow us to perform a first analyses of differences in decoding performance in broadly-defined functional regions of the brain, this analysis suffers from the limitations that there is no universally agreed upon functional atlas of the human brain, the exact location of functional regions might also depend on the task, and there is substantial between-subject variability (Bohland et al., 2009; Salehi et al., 2020). In the future we plan to address these shortcomings by leveraging a more bottom-up approach in the form of searchlight analyses.

Table 2: Regions that were included in the high-level visual ROI

| ID | Label | Names |
|----|-------|-------|
| 21 | L G_oc-temp_lat-fusifor | Lateral occipito-temporal gyrus (fusiform gyrus, O4-T4) |
| 21 | R G_oc-temp_lat-fusifor | Lateral occipito-temporal gyrus (fusiform gyrus, O4-T4) |
| 23 | L G_oc-temp_med-Parahip | Parahippocampal gyrus, parahippocampal part of the medial occipito-temporal gyrus, (T5) |
| 23 | R G_oc-temp_med-Parahip | Parahippocampal gyrus, parahippocampal part of the medial occipito-temporal gyrus, (T5) |
| 61 | L S_oc-temp_med_and_Lingual | Medial occipito-temporal sulcus (collateral sulcus) and lingual sulcus |
| 61 | R S_oc-temp_med_and_Lingual | Medial occipito-temporal sulcus (collateral sulcus) and lingual sulcus |
| 60 | L S_oc-temp_lat | Lateral occipito-temporal sulcus |
| 60 | R S_oc-temp_lat | Lateral occipito-temporal sulcus |
| 37 | L G_temporal_inf | Inferior temporal gyrus (T3) |
| 38 | L G_temporal_middle | Middle temporal gyrus (T2) |
| 72 | L S_temporal_inf | Inferior temporal sulcus |
| 37 | R G_temporal_inf | Inferior temporal gyrus (T3) |
| 38 | R G_temporal_middle | Middle temporal gyrus (T2) |
| 72 | R S_temporal_inf | Inferior temporal sulcus |

Table 3: Regions that were included in the low-level visual ROI

| ID | Label | Names |
|---|---|---|
| 2 | L G_and_S_occipital_inf | Inferior occipital gyrus (O3) and sulcus |
| 19 | L G_occipital_middle | Middle occipital gyrus (O2, lateral occipital gyrus) |
| 20 | L G_occipital_sup | Superior occipital gyrus (O1) |
| 42 | L Pole_occipital | Occipital pole |
| 57 | L S_oc_middle_and_Lunatus | Middle occipital sulcus and lunatus sulcus |
| 58 | L S_oc_sup_and_transversal | Superior occipital sulcus and transverse occipital sulcus |
| 59 | L S_occipital_ant | Anterior occipital sulcus and preoccipital notch (temporo-occipital incisure) |
| 65 | L S_parieto_occipital | Parieto-occipital sulcus (or fissure) |
| 2 | R G_and_S_occipital_inf | Inferior occipital gyrus (O3) and sulcus |
| 19 | R G_occipital_middle | Middle occipital gyrus (O2, lateral occipital gyrus) |
| 20 | R G_occipital_sup | Superior occipital gyrus (O1) |
| 42 | R Pole_occipital | Occipital pole |
| 57 | R S_oc_middle_and_Lunatus | Middle occipital sulcus and lunatus sulcus |
| 58 | R S_oc_sup_and_transversal | Superior occipital sulcus and transverse occipital sulcus |
| 59 | R S_occipital_ant | Anterior occipital sulcus and preoccipital notch (temporo-occipital incisure) |
| 65 | R S_parieto_occipital | Parieto-occipital sulcus (or fissure) |
| 22 | L G_oc-temp_med-Lingual | Lingual gyrus, ligual part of the medial occipito-temporal gyrus |
| 22 | R G_oc-temp_med-Lingual | Lingual gyrus, ligual part of the medial occipito-temporal gyrus |

Table 4: Regions that were included in the language ROI

| ID | Label | Names |
|---|---|---|
| 12 | L G_front_inf-Opercular | Opercular part of the inferior frontal gyrus |
| 13 | L G_front_inf-Orbital | Orbital part of the inferior frontal gyrus |
| 14 | L G_front_inf-Triangul | Triangular part of the inferior frontal gyrus |
| 25 | L G_pariet_inf-Angular | Angular gyrus |
| 15 | L G_front_middle | Middle frontal gyrus (F2) |
| 34 | L G_temp_sup-Lateral | Lateral aspect of the superior temporal gyrus |
| 36 | L G_temp_sup-Plan_tempo | Planum temporale or temporal plane of the superior temporal gyrus |
| 35 | L G_temp_sup-Plan_polar | Planum polare of the superior temporal gyrus |
| 4 | L G_and_S_subcentral | Subcentral gyrus (central operculum) and sulci |
| 26 | L G_pariet_inf-Supramar | Supramarginal gyrus |
| 9 | L G_cingul-Post-dorsal | Posterior-dorsal part of the cingulate gyrus (dPCC) |
| 10 | L G_cingul-Post-ventral | Posterior-ventral part of the cingulate gyrus (vPCC, isthmus of the cingulate gyrus) |

## A.5 PER-SUBJECT RESULTS

Results for individual subjects can be found in Figure 6. Among all subjects, we found similar converging results for decoding accuracies when comparing models, feature modalities, and modality-agnostic with modality-specific decoders.

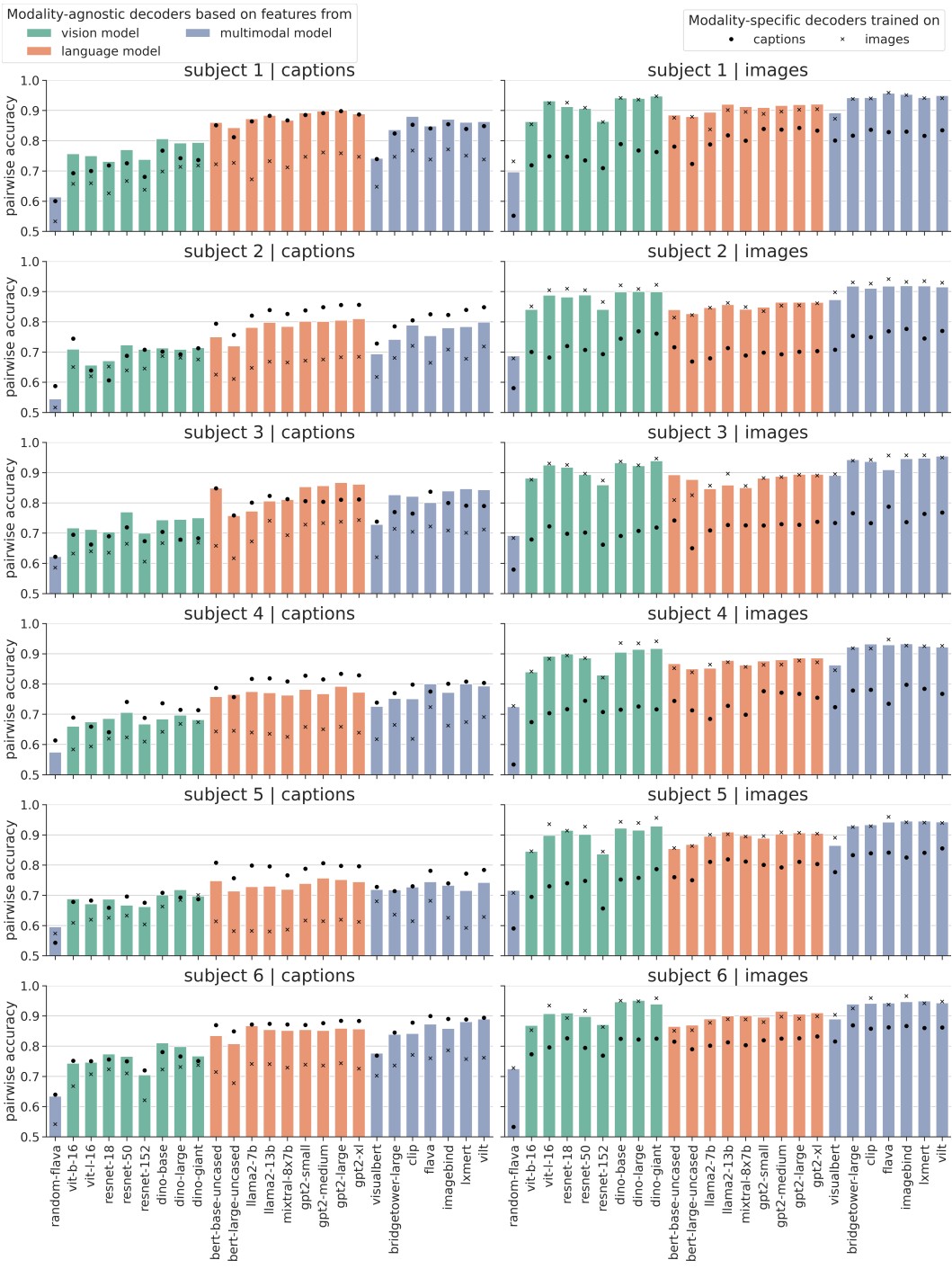

Figure 6: Pairwise accuracy per subject

