# OpenReview forum: "Modality-Agnostic fMRI Decoding of Vision and Language"
_ICLR.cc/2024/Workshop/Re-Align — ICLR 2024 Workshop Re-Align Poster_

### Official Review · Reviewer_H4KE · 2024-02-23
**fMRI Decoding Across Modalities**

**Rating:** 2
**Fit:** 3
**Confidence:** 3

**Workshop Review:**

This work introduces a novel large-scale fMRI dataset of subjects looking at images or text, and demonstrates how decoders can be trained using this dataset to map brain activity onto stimulus representations using pre-trained vision/language models. Their decoders (1) perform as well as modality-specific decoders despite added stimulus modality uncertainty, (2) perform just as well mapping onto unimodal model representations as multimodal representations, (3) high-level visual regions perform well on text and images whereas language and low-level visual regions do best at language and text respectively.

I applaud the authors on intending to release their new dataset. This will be a valuable asset to the broader neuroimaging community.

The authors conclude that modality-agnostic decoders achieve performance as good as the modality-specific decoders despite having to overcome the challenge of not knowing the modality of the stimulus. However, a challenge that the modality-specific decoders had to overcome (in comparison to the modality-agnostic decoders) was being trained with half the data, right? If you were to keep the total number of training samples the same, would modality-agnostic and modality-specific decoder performance still be comparable?

Could you clarify exactly how pairwise accuracy was calculated? E.g., for decoding accuracy for captions, did you take the 70 test caption stimuli, convert them to the latents of the respective models, and then check whether the cosine similarity was higher between the ground truth caption and the predicted latent (from the ridge model) than each of the 69 other caption stimuli? AKA chance performance = 0.5?

Do you think your results would generalize to non-linear decoding models? E.g., using ridge regression to the concatenation of the image and text latent representations may limit impairments caused from uncertainty in input stimulus modality, whereas non-linear neural networks (which is the current approach used for SOTA decoding models) might struggle more.

Could add references to relevant fMRI image reconstruction papers:

Scotti, P., Banerjee, A., Goode, J., Shabalin, S., Nguyen, A., Dempster, A., ... & Abraham, T. (2023). Reconstructing the mind's eye: fMRI-to-image with contrastive learning and diffusion priors. Advances in Neural Information Processing Systems, 36.

Chen, Z., Qing, J., Xiang, T., Yue, W. L., & Zhou, J. H. (2023). Seeing beyond the brain: Conditional diffusion model with sparse masked modeling for vision decoding. In Proceedings of the IEEE/CVF Conference on Computer Vision and Pattern Recognition (pp. 22710-22720).

**Reason For Not Giving Higher Score:**

Impact is not very strong (modality-specific decoders doing as well as modality-agnostic decoders using linear models), perhaps not wide enough interest for highest score.

**Reason For Not Giving Lower Score:**

Well-written paper, concrete results, and releases a new fMRI dataset.

**Reviewer Domain:**

cognitive science

---

### Official Review · Reviewer_SvB2 · 2024-02-23
**Great submission**

**Rating:** 2
**Fit:** 3
**Confidence:** 3

**Workshop Review:**

This paper is taking a step to addressing modality-independent decoding of fMRI data, which is a very important question. The authors introduces an fMRI dataset and made some initial analysis using modality-agnostic decoders as well as modality-specific decoders. Overall the paper is well written, the idea is novel, and the findings are interesting. I would recommend acceptance, but I have a few concerns:
- The language ROI seems to include a lot of regions that are not language specific, such as the superior frontal gyrus, which isn’t usually considered a language region but more like a high-level cognitive region (e.g., Friederici, 2011). I could see the definition of high-level visual area being somewhat controversial as well. Could the authors explain why the three ROIs are defined as it is? Alternatively, it would be ideal if the authors could base this analysis on smaller individual brain parcels and obtain whole-brain maps where each parcel shows its respective accuracy.
- The decoding results in 3.1 only included some general observations, but to claim (non-)significance the authors should include the stats for significance tests.
- In the appendix the authors mentioned “Preliminary experiments on a subset of models showed that these representations allow to build better decoders than representations extracted from [CLS] tokens.” This is interesting and I’d hope to see what exactly the results look like (e.g., how much of an improvement there is when using the average rather than the [CLS]).

Additionally, I also have a few suggestions for making the paper clearer:
- It would be clarifying if the authors provides an overview on how the decoding pipeline works in the main text. In the interest of space, I think some detailed parameters of fMRI scanning and preprocessing may be moved to appendix without affecting clarity.
- Similarly, it would be clarifying if the authors define early on in the paper what exactly they mean by “decoding”/“decoders” in this context. For example, here the authors are doing an image/text retrieval based on similarity scores, but in other contexts “decoding” could mean using a generative model to reconstruct the original stimuli. This isn’t made clear until the end of the method section, but it would have been helpful to know this information earlier in introduction/abstract.
- The authors mentioned that “this new fMRI dataset will be released publicly in an upcoming publication”. It would be helpful if in the camera-ready version the authors could link to that paper (or a preprint).
- It could be helpful if the authors could discuss a bit more on how this dataset may be used by others.

**Reason For Not Giving Higher Score:**

The paper could be written more clearly, and there are a few issues that need to be addressed (see my comments above).

**Reason For Not Giving Lower Score:**

The paper presents a great dataset and methods for investigating multimodal representations in the human brain, and it fits the topic of the workshop.

**Reviewer Domain:**

neuroscience

---

### Decision · Program_Chairs · 2024-03-02

Accept (Poster)